# THE ROLE OF DATA IN MODEL MERGING

## ABSTRACT

Model merging procedures often include components that are data-dependent, but the effect of data is often overlooked. Focusing on two key components of the merging process – the computation of permutation symmetries and the correction of activation statistics, we study how the amount and difficulty of data affects model merging. Our experiments show that choice of data significantly influences merged model performance, with suboptimal choices resulting in up to $2\times$ worse performance than the ideal. We also demonstrate that data affects merged model performance primarily through the correction of activation statistics and that skewed data subsets consistently lead to incorrect estimates of these statistics.

## 1 INTRODUCTION

Averaging neural network weights, or model merging, has recently attracted significant attention for its utility in both scientific and practical settings. Past work has largely focused on algorithmic improvements to merging methods (Stoica et al., 2023; Xu et al., 2024). However, many of the components of merging procedures, such as neuron permutation matching (Ainsworth et al., 2022), correction of activation statistics(Jordan et al., 2022), or additional optimization (Nasery et al., 2025), are also data dependent. Understanding how data affects merging could be valuable in situations where some or all of the original data is inaccessible (Nasery et al., 2025), the dataset is simply too large to be used in its entirety (Verma & Elbayad, 2024), or when handling different datasets in multi-task settings (Lasby et al., 2025).

In this work, we focus on a simple two-step merging procedure that is often used to merge independently-trained networks: (1) activation matching (Li et al., 2015; Ainsworth et al., 2022), which finds a permutation symmetry that brings models into the same loss basin, allowing them to be merged and, (2) correcting the activation statistics of the merged model via REPAIR, which dramatically improves its performance (Jordan et al., 2022). Both steps are data-dependent but have very different purposes – analyzing this setting is a good first step towards a holistic understanding of the effect of data on merging.

We study the effect of data on the performance of the merged model along two axes: (a) the amount of data and, (b) the type of data – specifically, example difficulty. Our main findings are summarized as follows:

- The choice of data used for model merging significantly affects the performance of the merged model – in our experiments, models merged using suboptimal subsets of data perform up to $2\times$ worse than the ideal.
- While both steps of the model merging framework are influenced by the choice of data, we find that it is significantly more critical for the correction of activation statistics.
- Finally, we analyze the interaction between data, activation statistics, and merged model performance from the perspective of variance collapse (Jordan et al., 2022). We show that skewed subsets of data lead to incorrect estimates of activation statistics, leading to worse performance.

## 2 THE EFFECT OF DATA ON MODEL MERGING

We independently train networks and merge them using a two-step procedure: (1) first, we use activation matching to compute a permutation symmetry and then, (2) we use REPAIR to correct the

activation statistics of the merged model. We refer the reader to Appendix B and Appendix C for details. Results for ResNet20s (He et al., 2016) with BatchNorm (Ioffe & Szegedy, 2015) trained on CIFAR-10 (Krizhevsky et al., 2009) are shown in Figure 1; additional results for ResNet20s with LayerNorm (Ba et al., 2016) and ResNet50 models trained on ImageNet (Deng et al., 2009) are presented in subsection D.2 and subsection D.4 respectively. We use the *accuracy barrier* – the difference between the average accuracy of the endpoint models and the accuracy of the merged model – to evaluate merged models. Ideal merging corresponds to zero barrier.

To understand the overall effect of data on model merging, we vary: (a) amount of data, and (b) example difficulty used for model merging. We decide to employ the notion of example difficulty for two reasons: it provides a convenient yet meaningful way to categorize data and, in simpler model merging settings, its role in illustrating the effect of data has been significant (Iyer et al., 2024) – for instance, Paul et al. (2021) demonstrates that the error of the averaged models is significantly different on dataset examples that differ in difficulty. We measure example difficulty using the EL2N score (Paul et al., 2021).

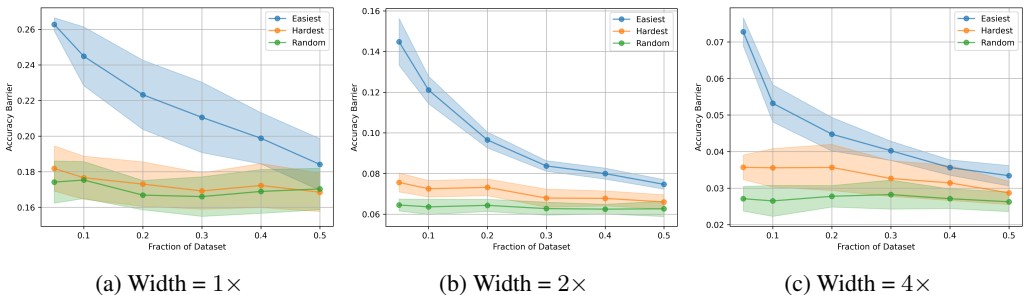

(a) Width = $1\times$        (b) Width = $2\times$        (c) Width = $4\times$

Figure 1: Accuracy barriers as a function of the amount of data used for model merging, for different example difficulty levels. We show results averaged over three pairs of networks, with error-bars showing standard deviation. The same data is used for both activation matching and REPAIR. Random data yields the smallest barriers (i.e. best merging) with a small amount of data while easy data yields much larger barriers, even when a large amount of such data is used.

We find that the choice of data can have a dramatic effect on the accuracy barriers of merged models. Using random data yields the smallest barriers, hard data yields slightly higher barriers, and using easy data leads to significantly larger accuracy barriers – when only a small amount of data is available, easy data can lead to $2\times$ the barrier compared to using an equivalent amount of random data. In fact, even using $50\%$ of the easiest data yields a higher accuracy barrier than using just $5\%$ of hard or random data. While using more data can yield smaller accuracy barriers (especially when using easy data), this generally only holds up to a limit – we see that after using $\approx 30\%$ of the hardest data, our accuracy barriers largely remain constant. In fact, using even $5\%$ of random data is sufficient to achieve the smallest accuracy barriers, especially in wider networks.

These results establish that the choice of data is an important consideration, but it is still not clear *why* merging is sensitive to the amount and kind of data used. As a reminder, the model merging framework we employ consists of two parts – finding the permutation symmetry via activation matching, and then correcting the activation statistics via REPAIR. Activation matching finds the permutation symmetry that maximizes the correlation between the activations of the models being matched. On the other hand, REPAIR corrects activation statistics by setting the statistics of the merged model to the mean of the endpoint model statistics.

Given that both steps are data-dependent but have distinct purposes, how does data impact them individually and how does it reflect in accuracy barriers? In the following subsections, we answer this question by isolating and evaluating each step using subsets of data of different sizes and difficulties.

**Effect through Activation Matching** In Figure 2, we vary the data used for activation matching while using the entire dataset for REPAIR. We find that while our observations from Figure 1 still hold to some extend, they are much less pronounced, especially for narrow models. As the amount of data used becomes larger, it becomes harder to see the influence of example difficulty on accuracy barriers.

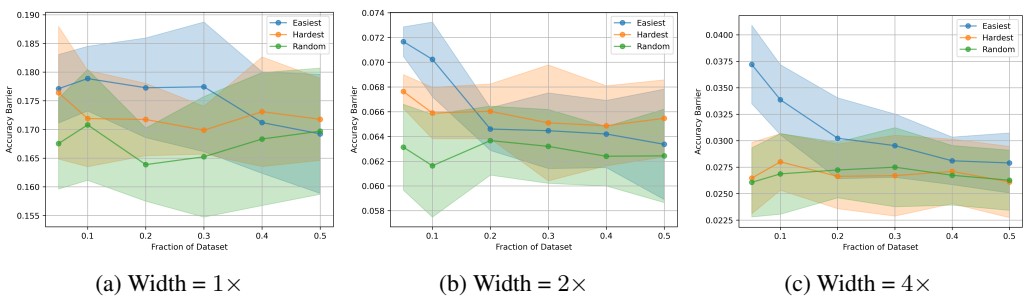

(a) Width = $1\times$             (b) Width = $2\times$             (c) Width = $4\times$

Figure 2: Accuracy barriers versus amount of data, when data for activation matching is varied and the full training dataset is used for REPAIR.

**Effect through REPAIR** In Figure 3, we use the entire training dataset for activation matching while varying the data for REPAIR. We once again observe striking differences in barriers as the amount of data and difficulty are changed, suggesting that choice of data affects barriers more through the activation statistics than through permutation symmetries – indeed, note how Figure 1 and Figure 3 are nearly identical.

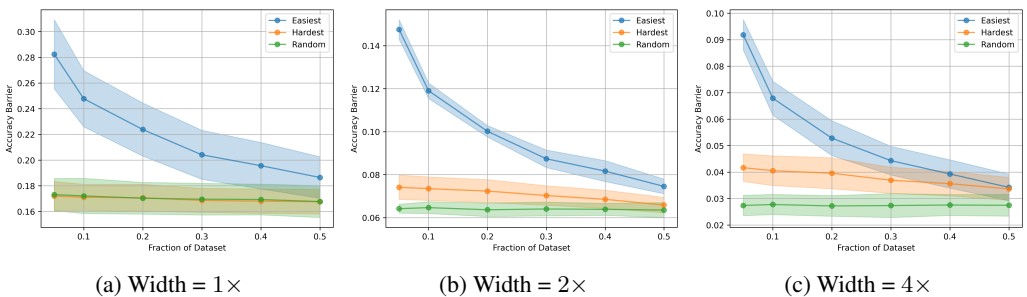

(a) Width = $1\times$             (b) Width = $2\times$             (c) Width = $4\times$

Figure 3: Accuracy barriers versus amount of data, when the full training dataset is used for activation matching and data is varied for REPAIR.

Overall, our experiments demonstrate that while data influences both the permutation symmetry and the correction of activation statistics, it is the effect on the latter that dominates. In the following section, we analyze this interplay between data, activation statistics and performance barrier.

## 3  DATA AND VARIANCE COLLAPSE

So far, our results show that data *primarily* affects accuracy barriers through the (re-)computation of activation statistics via REPAIR. Jordan et al. (2022) show that a major contributor to accuracy barriers is "variance collapse" – a sharp decrease in the variance of activations in the merged model relative to the endpoint models, especially in the deeper layers of the model – and propose RE-PAIR to mitigate it. Can we also understand the interactions between accuracy barriers, activation statistics, and example difficulty through the same lens?

Indeed, we find that we can – in Figure 4, we plot the barrier of a merged model against the variance ratio averaged over all layers, for different choices of data used for REPAIR. The variance ratio measures the ratio of the variance of activations of the merged model and endpoint models – refer to Appendix B for more details. In (a), we use the same data subsets for both activation matching and REPAIR as in Figure 1, and see a striking correspondence between example difficulty and the variance ratio. Correcting activation statistics using easy or hard data leads to consistently incorrect estimates of the activation variance, with estimates becoming more accurate as more data is used. In contrast, using even a small amount of random data yields roughly the same activation variance and barriers. Subfigure (b) is in the same setting as Figure 2, where activation matching is done on different subsets of the data while REPAIR is done using the entire dataset. Here, the results are starkly different, as the strong correspondence between example difficulty and variance ratio

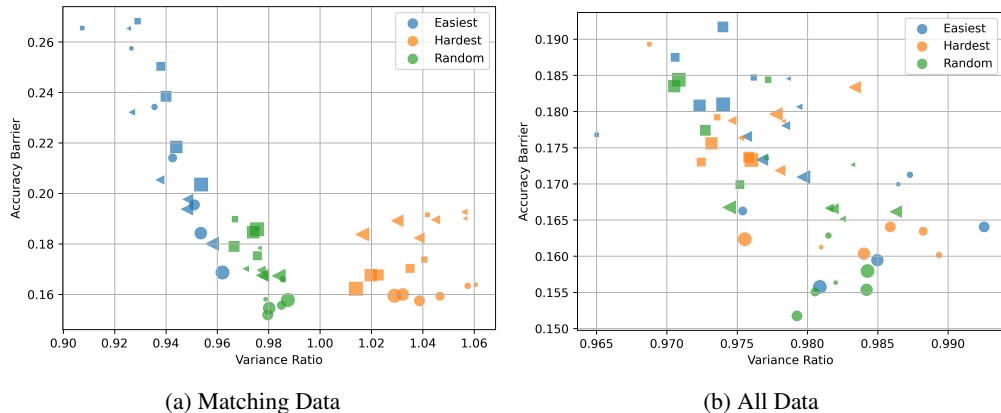

(a) Matching Data  (b) All Data

Figure 4: Plot showing the relationship between accuracy barriers and variance ratios when (a) the same data is used for activation matching and REPAIR and (b) data for activation matching is varied, while all the data is used for REPAIR. Theoretically, a variance ratio of exactly 1 is optimal, which REPAIR strives to do. Each point corresponds to a merged model, with unique markers indicating a unique pair of endpoint networks, and colors indicating example difficulty. The marker size corresponds to the amount of data used for activation matching and/or REPAIR.

vanishes. Once again, we provide complementary results for wider networks in subsection D.1, for LayerNorm networks in subsection D.2, and for ResNet50s on ImageNet in subsection D.4.

The above results also mirror what we see in section 2 – using a small amount of easy data generally results in a variance that is too low, leading to larger accuracy barriers. As the amount of data used is increased, the estimate becomes more accurate and performance improves. Furthermore, when all the data is used and the correspondence breaks, different data subsets yield similar performance.

In addition to reinforcing our previous observations, the results presented here help us further understand how data impacts the activation statistics of models. It also suggests that example difficulty also captures example-level differences in activations – and as a consequence, choosing a skewed data subset could significantly impact the performance of the merged model.

## 4  CONCLUSION AND FUTURE WORK

This paper investigates how data affects model merging by varying the amount and difficulty of the data used for model merging. We show that choice of data can have a significant effect on merged model performance – using easy data often yields suboptimal barriers while relatively small amounts of randomly sampled data results in low barriers. We also find this difference is primarily due to the effect of data on activation statistics. Finally, we analyze this further through the lens of variance collapse, and demonstrate that skewed data affects the activation variance ratio in distinct ways.

We view our work as an initial step toward understanding the effect of data on model merging. At the same time, it also raises many compelling questions for future work to address. Data affects different parts of the merging pipeline to different extents – for instance, we see that the effectiveness of activation matching is relatively insensitive to data compared to REPAIR. Is this difference due to the stability of feature learning or just due to the suboptimality of activation matching? It is also worth considering if and how our observations would change when other merging methods (which possibly contain data-dependent components) are used, in the context of atypical data distributions or in a practical setting like multi-task learning. Finally, we hope that a better understanding of how different components of the merging pipeline are affected by the choice of data can lead to practical, automated data curation methods for the situations where the ideal data for merging is either not known or not available.

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

## A    RELATED WORK

**Weight Space Permutation Symmetries.**    Recently, the permutation symmetries of neural network weight spaces has been a topic of great interest (Simsek et al., 2021; Brea et al., 2019), with many works proposing methods that can find layer-wise permutations that can "align" a network to a reference network – by which we mean that the permuted network lies in the same loss basin as the target network (Ainsworth et al., 2022; Singh & Jaggi, 2020; Jordan et al., 2022). The significant success of these approaches has led to the hypothesis that when permutation symmetries are accounted for, independently initialized SGD solutions have no error barrier between them (Entezari et al., 2021). While it has been theoretically shown that this conjecture is false (see Appendix A.6

of Ainsworth et al. (2022)), it is unclear whether this holds for the larger networks used in practice. Despite the progress, zero error barriers between independently trained networks have only been observed in very wide networks – while this is partially attributed to the algorithmic difficulty of finding these permutations in the first place, it obfuscates the answer even further.

**Linear Mode Connectivity.** Closely related to the topic of permutation symmetries is linear mode connectivity (LMC) Frankle et al. (2020) – the observation that network weights trained on different sets of SGD noise (i.e. different batch orders, augmentations etc.) can be linearly interpolated to yield a distinct network that achieves low loss similar to the parent networks. This observation, in general, holds under two strong conditions: (a) the two parent networks have the same initialization, and (b) the two networks undergo shared training for a small number of epochs before being trained on distinct sets of SGD noise. Indeed, the goal of aligning trained networks with the help of permutations is to induce linear mode connectivity without the need for these strong conditions. It is also important to note that *mode connectivity* has been observed between independently trained networks – their weights can be connected by simple or piecewise linear curves such that the loss along this curve remains small Draxler et al. (2018); Garipov et al. (2018). Mirzadeh et al. (2020) observe LMC in the context of multitask and continual learning. Zhou et al. (2023) introduce the notion of Layerwise Linear Feature Connectivity (LLFC), which extends the notion of LMC to activations, claiming that this is, in fact, a more general form of LMC.

**Weight Averaging and Model Merging.** Weight averaging and model merging are gaining traction as a practical way to improve the performance and capabilities of neural networks, especially in the current era of very large models, where training them from scratch is exceedingly prohibitive. Wortsman et al. (2022) shows that models finetuned with different hyperparameter configurations (from the same pretrained based model) lie in the same basin in the loss landscape and that averaging them can improve performance and robustness. Ilharco et al. (2022) introduces the concept of "task-vectors", which specify directions that correspond to individual tasks and show that one can add and subtract these vectors to add or remove specific capabilities from models. Yadav et al. (2023) investigates sources of "interference" in models to be merged, and improves the model merging paradigm by resolving these interferences and reducing the loss of relevant information which occurs as a result of averaging weights.

**Example Importance.** Paul et al. (2021) introduces the Gradient Normed (GraNd) and Error L2-Norm (EL2N) scores, which identify important examples in a dataset early in training, across different architectures and training configurations. In our experiments, we will make use of the EL2N score in order to form subsets of data based on difficulty. They also show that networks become "stable to SGD noise" (in the LMC sense) with respect to easier examples early in training, while this happens much later in training for harder examples – this further motivates the big picture idea for this project. Toneva et al. (2018) shows that "easy" examples are generally learned earlier in training and are rarely forgotten. Furthermore, these examples do not contribute significantly to final generalization performance. Baldock et al. (2021) propose the notion of "effective prediction depth" to measure the difficulty of an example – the number of layers required to determine the class prediction on that example. Kwok et al. (2024) do an extensive, systematic analysis of different measures of example difficulty, finding that they are generally correlated on average (albeit noisy over individual runs), and use their findings to "fingerprint" model architectures using a small number of sensitive examples.

# B PRELIMINARIES

## B.1 PERMUTATION SYMMETRIES AND ACTIVATION MATCHING

One can apply arbitrary layerwise permutations to the weight vectors of a neural network and preserve the function it expresses, as long as the following layer is also accordingly permuted.

More formally, consider an L-layer fully-connected network such that

$$f(x, \Theta) = z_{L+1}, \quad z_{l+1} = \sigma(W_l z_l + b_l), \quad z_1 = x$$

Then, consider the following way of rewriting $z_{l+1}$

$$z_{l+1} = P^{\mathrm{T}} P z_{l+1} = P^{\mathrm{T}} P \sigma(W_l z_l + b_l) = P^{\mathrm{T}} \sigma(P W_l z_l + P b_l)$$

where $P$ is some permutation matrix. That is, if we reorder the weights of layer $l+1$ according to $P^{\mathrm{T}}$, the result is functionally equivalent model weights $\Theta' = \Theta$ *except*

$$W_l' = P W_l, \quad b_l' = P b_l, \quad W_{l+1}' = W_{l+1} P^{\mathrm{T}}$$

Thus, the goal is to find a permutation $P$ for each layer which minimizes some measure of distance between the two networks to be aligned. In this work, we will primarily focus on *activation matching*, which attempts to minimize the squared distance between the layer-wise activations of the two networks (A and B) to be aligned, as follows:

$$P_l = \arg\min_{P} \sum_{i=1}^{n} ||Z_{:,i}^{(A)} - P Z_{:,i}^{(B)}||_2$$

where $Z^{(A)}$ and $Z^{(B)}$ denote the layer-activations of the respective networks, and $i$ iterates over the data samples. The latter is important to note – we want to control what data samples we compute our permutations with respect to, and this makes activation matching an ideal choice for the experiments we plan on performing.

While the analysis above is performed assuming a fully-connected network, it can be generalized to other architectures, such as convolutional networks, in a fairly straightforward manner.

There exist other formulations and implementations of activation matching – for instance, Li et al. (2015) implements activation matching by *maximizing the sum of the correlations* $\sum \mathrm{corr}(Z_{:,i}^{(A)}, P Z_{:,i}^{(B)})$. Furthermore, prior work indicates that there are two ways to solve such problems: by solving a linear sum assignment problem via the Hungarian algorithm to maximize a similarity (which is what we use), or by framing it as an ordinary least squares (OLS) regression task to minimize a cost, as in **?**. While we do not consider the alternative formulations and choices in this work, it is worth considering what the impact of these choices is on the permutations found by the algorithm, even though the overall objectives are fairly similar.

## B.2 REPAIR

Introduced by Jordan et al. (2022), REPAIR aims to reduce the accuracy barrier between the endpoint models and merged model by addressing "variance collapse", which is a sharp decrease in the variance of the activations of the merged model relative to the endpoint models.

More formally, for some layer $l$, let $X_1$ and $X_2$ be the preactivations of the two endpoint models, and $X_a$ be the preactivations of the averaged model. Let $v_1, v_2,$ and $v_a$ be the sum of the variance of activations, across every neuron for each network. Then, the variance collapse for the given layer is measured via the quantity $\frac{2v_a}{v_1+v_2}$, which we call the *variance ratio*.

In practice, REPAIR is performed by introducing BatchNorm layers to the endpoint models and merged model. Using the BatchNorm endpoint networks, one can then compute the correct statistics for the endpoint models, which can then be used to compute the same for the merged model. Specifically, we want that:

$$\mathbb{E}[X_a] = (\mathbb{E}[X_1] + \mathbb{E}[X_2])/2$$
$$\mathrm{std}[X_a] = (\mathrm{std}[X_1] + \mathrm{std}[X_2])/2$$

## B.3 EXAMPLE DIFFICULTY AND THE EL2N SCORE

Example difficulty (or importance) aims to identify and understand the impact of individual data samples on the generalization of a model. Samples that are more influential are generally difficult

to learn and vice-versa. One such example difficulty metric is the EL2N score (**?**)), which assigns a score to each example in a training dataset – the higher the score, the more difficult/important an example is. Formally, the EL2N score of a training sample $(x.y)$ is defined as:

$$\text{EL2N}(x, y) = \mathbb{E}||f(w_t, x) - y||_2$$

where $f(w_t, x)$ is the output of the neural network in the form of a probability vector (e.g. after the application of a softmax) using the weights $w$ at some training iteration/epoch $t$. The expectation is taken over training runs with different initializations. Averaging the EL2N score over multiple initializations of a neural network architecture results in a relatively consistent ordering of data samples in a given dataset – the higher the EL2N score, the more "important" or "difficult" an example is deemed to be.

## C  METHODOLOGY AND EXPERIMENTAL SETUP

We train ResNet20 models on CIFAR-10 with SGD with momentum = 0.9 and weight decay = 1e-4, using a batch size of 128 and cross-entropy loss. We use a base learning rate of 0.4 and 0.1 for BatchNorm and LayerNorm respectively, starting with a linear warmup and then decaying the learning rate using a cosine annealing scheduler. Models are trained for a total of 200 epochs.

Our model merging procedure consists of two main steps:

1. First, use activation matching to compute the permutation symmetry that aligns the two independently trained networks in weight space. Then, we average the two aligned networks in weight space to obtain the merged model.

2. Next, we correct the normalization statistics of the merged model using REPAIR. When correcting or computing normalization statistics, we employ a larger batch size of 1000.

When doing activation matching or REPAIR, we sample data exclusively from the training set, while all accuracy barriers are computed on the test set.

To evaluate the performance of a merged model, we use the notion of an accuracy barrier. Let the absolute accuracy of the endpoint models be $P_1$ and $P_2$, and the accuracy of the merged model be $P_a$. Then the accuracy barrier is the quantity $\frac{P_1+P_2}{2} - P_a$.

To measure example difficulty, we use the EL2N score with small and large scores corresponding to easy and hard examples respectively – specifically, we take the mean of the EL2N scores at training epoch 10, across 10 different ResNet20 networks trained. These models were trained with similar hyperparameter configurations to those stated above.

The activations used for variance ratios are computed on a random 25% subset of the training dataset.

## D  ADDITIONAL RESULTS

### D.1  ADDITIONAL RESULTS FOR BATCHNORM NETWORKS

### D.2  RESULTS FOR LAYERNORM NETWORKS

### D.3  RESULTS WHEN REPAIR IS NOT USED

### D.4  RESULTS FOR RESNET50S ON IMAGENET

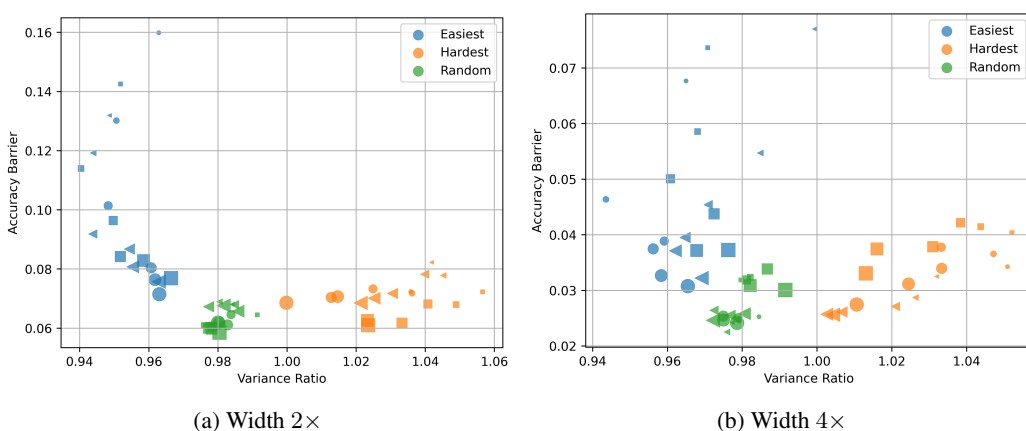

(a) Width 2×

(b) Width 4×

Figure 5: Accuracy barriers versus variance ratio for BatchNorm networks when the same data is used for both activation matching and REPAIR.

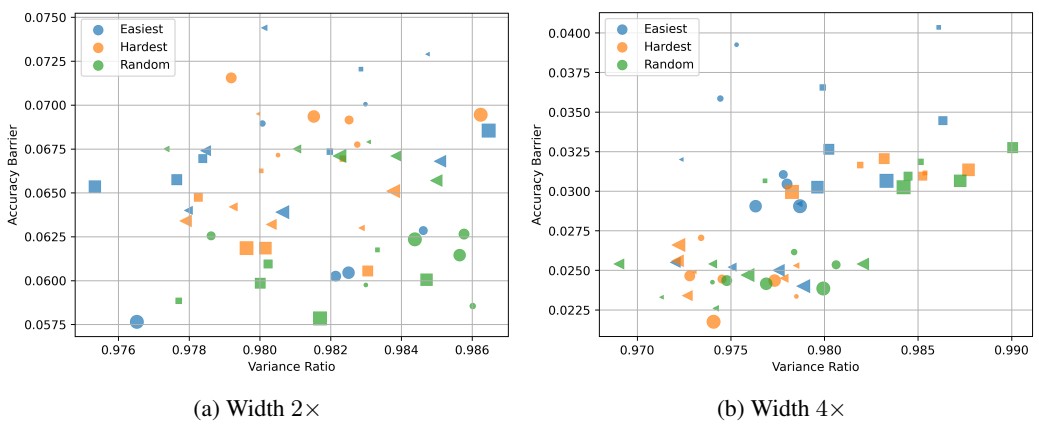

(a) Width 2×

(b) Width 4×

Figure 6: Accuracy barriers as a function of the amount of data used for model merging for Batch-Norm networks when data for activation matching is varied, but all the data is used for REPAIR.

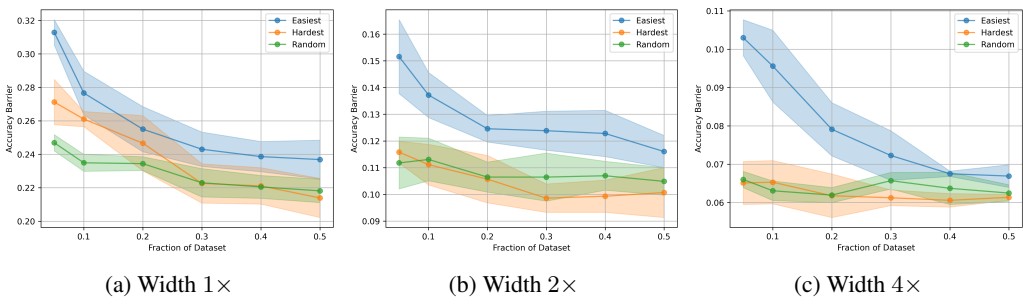

(a) Width 1×

(b) Width 2×

(c) Width 4×

Figure 7: Accuracy barriers as a function of the amount of data used for model merging for Layer-Norm networks when the same data is used for both activation matching and REPAIR.

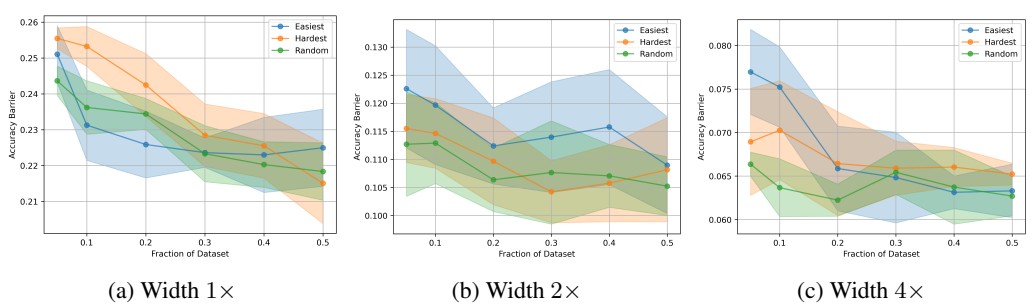

(a) Width 1×      (b) Width 2×      (c) Width 4×

Figure 8: Accuracy barriers as a function of the amount of data used for model merging for Layer-Norm networks when data for activation matching is varied, but all the data is used for REPAIR.

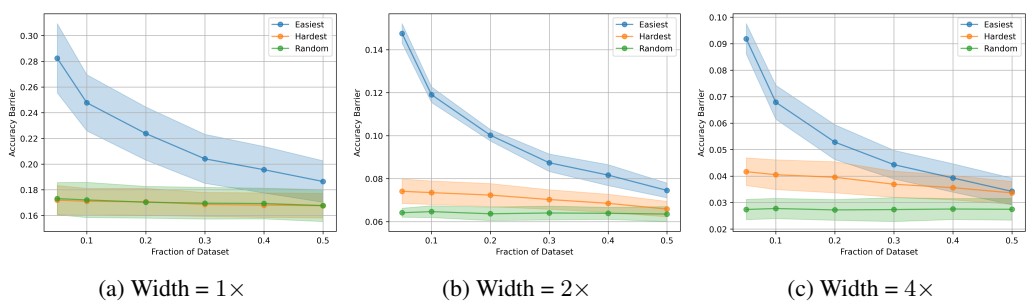

(a) Width = 1×      (b) Width = 2×      (c) Width = 4×

Figure 9: Accuracy barriers as a function of the amount of data used for model merging for Layer-Norm networks when all the data is used for activation matching, and the data used for REPAIRis varied.

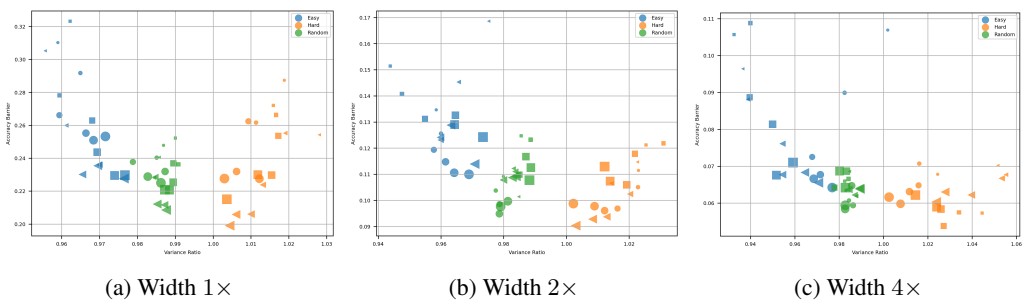

(a) Width 1×      (b) Width 2×      (c) Width 4×

Figure 10: Accuracy barriers versus variance ratio for LayerNorm networks when the same data is used for both activation matching and REPAIR.

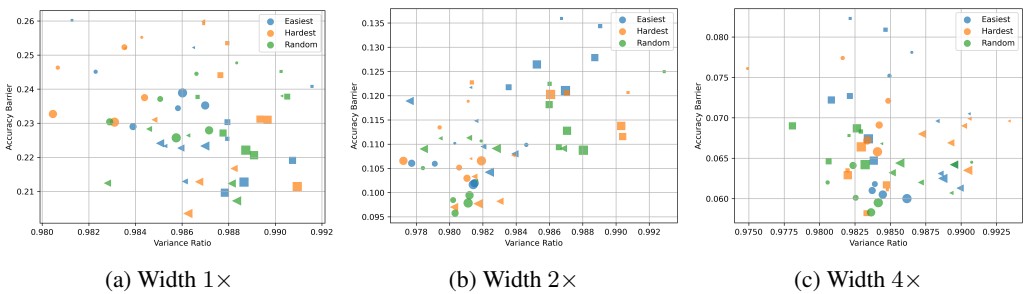

(a) Width 1×      (b) Width 2×      (c) Width 4×

Figure 11: Accuracy barriers as a function of the amount of data used for model merging for Layer-Norm networks when data for activation matching is varied, but all the data is used for REPAIR.

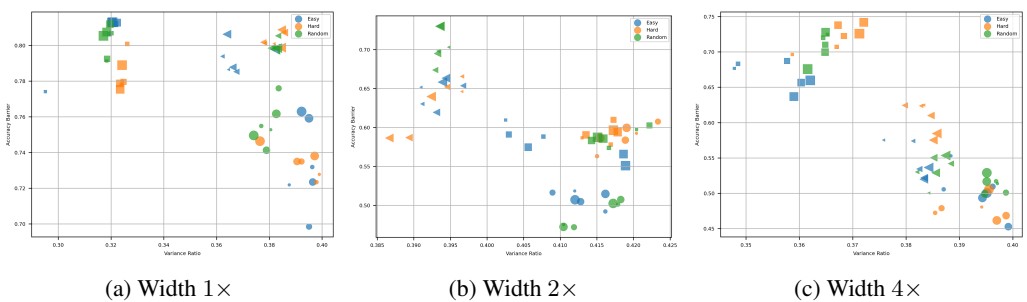

(a) Width 1×      (b) Width 2×      (c) Width 4×

Figure 12: Accuracy barriers as a function of the amount of data used for model merging for Batch-Norm networks when data for activation matching is varied, and activation statistics are left uncorrected.

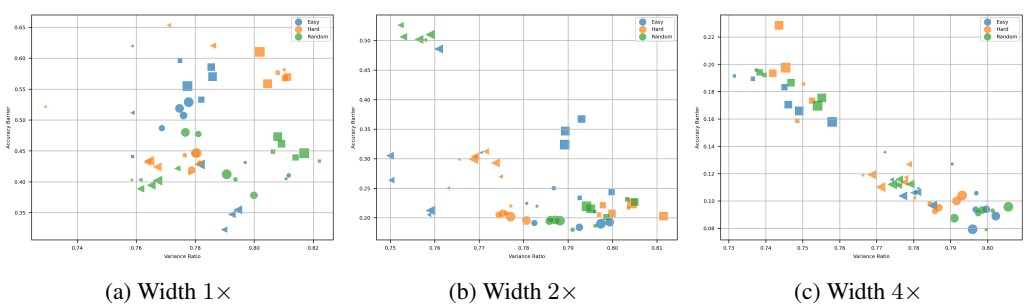

(a) Width 1×      (b) Width 2×      (c) Width 4×

Figure 13: Accuracy barriers as a function of the amount of data used for model merging for Layer-Norm networks when data for activation matching is varied, and activation statistics are left uncorrected.

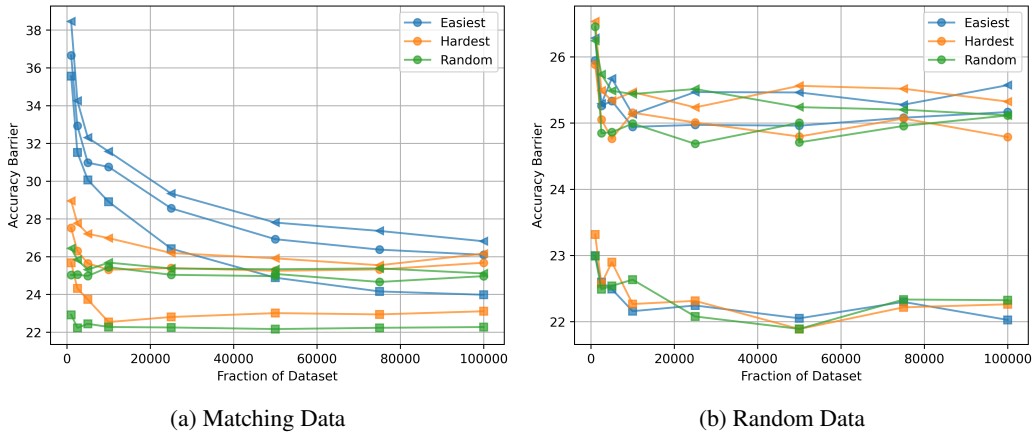

(a) Matching Data      (b) Random Data

Figure 14: Accuracy barriers as a function of the number of samples used for model merging for ResNet50 networks trainend on ImageNet. In (a), the same data is used for both activation matching and REPAIR, while in (b) we use an equivalent amount of randomly sampled data for REPAIR while varying data for activation matching with difficulty.

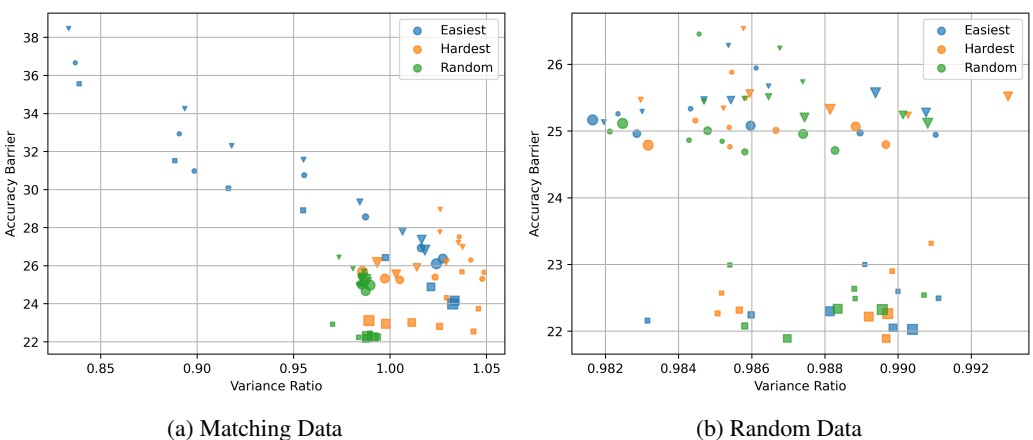

(a) Matching Data

(b) Random Data

Figure 15: Accuracy barriers as a function of the variance ratio for ResNet50 networks trained on ImageNet. In (a), the same data is used for both activation matching and REPAIR, while in (b) we use an equivalent amount of randomly sampled data for REPAIR while varying data for activation matching with difficulty.

