# OpenReview forum: "The Role of Data in Model Merging"
_ICLR.cc/2026/Workshop/Sci4DL — Sci4DL 2026_

### Official Review · Reviewer_HmxD · 2026-02-20

**Fit:** 3
**Significance:** 3
**Confidence:** 2

**Summary:**

Model Merging typically involve access to data for permutation matching and activation statistics corrections.
In real-world situations, it is often desirable or necessary to choose the data to use (either new, or a subset), rather than just using the original training data.

This paper investigates how quality of the merged model is affected by
1) The amount of data used for training;
2) the type of data used (""difficulty"" based on EL2N score)

Findings:
1) Sub-optimal choice of data can lead to very poor performance
2) activation statistics is the most critically affected phase of merging
3) In particular, skewed subsets of data are problematic

**Strengths:**

1) The abstract and claims appear to be well justified.
2) Looking at two distinct aspects of the merge-data is useful.
3) The experiments look well-reasoned and the paper is clearly explained.
4) Comparing the two different merging processes fits the Sci4DL ethos well.
5) It may well be that data being "easy" is actually a proxy for a more specific factor, but this paper identifies that as an area for further study, while already providing useful results
6) The appendices are useful and thorough

**Suggestions:**

1) Code availability: It would help in reproduction and extension of this work to have access to the code base used.
2) Terminology: I am not especially familiar with Model Merging, so this may be unnecessary, but it may be beneficial to indicate whether "accuracy barrier" is a measure that is being introduced in this paper, or a standard term of art. Could there be an explanation of why "ideal merging corresponds to zero barrier" rather than to the better of the two models?
3) Future Extension: Could more discussion be provided on why Hardest data seems to be a little worse than Random data (though both are better than Easiest.) Do you have results for "Mean Difficulty" tranche of data?
4) Further context: In your consideration of how difficult examples aid generalisation, the authors may find this paper interesting: Feldman, V., & Zhang, C. (2020). What Neural Networks Memorize and Why: Discovering the Long Tail via Influence Estimation (arXiv:2008.03703). arXiv. https://doi.org/10.48550/arXiv.2008.03703

---

### Official Review · Reviewer_Kzyz · 2026-02-24

**Fit:** 3
**Significance:** 3
**Confidence:** 3

**Summary:**

This paper investigates how data affects model merging performance, focusing on two components: (1) activation matching (finding permutation symmetries) and (2) REPAIR (correcting activation statistics). By varying the amount and difficulty of data used for merging, the authors show that:

1. Data choice significantly affects merged model performance (up to 2× worse with suboptimal choices)
2. Easy examples yield larger accuracy barriers; small amounts of random data work well
3. The effect operates primarily through REPAIR, not activation matching
4. Variance collapse explains the mechanism: skewed data leads to incorrect variance estimates

Experiments span ResNet20 on CIFAR-10 (BatchNorm and LayerNorm) and ResNet50 on ImageNet.

**Strengths:**

### Clear, Focused Investigation
The paper isolates a single, often-overlooked factor in model merging: the choice of data. While prior work has focused on algorithmic improvements, this paper asks a complementary question: what data should we use? This is practically relevant when original training data is unavailable or too large.

### Clean Experimental Design
The methodology is well-structured:
- Vary data along two meaningful axes (amount and difficulty)
- Isolate the two merging steps (activation matching vs. REPAIR) to determine where data matters most
- Use variance ratio as an interpretable diagnostic linking data to performance

The comparison between Figures 1-3 cleanly demonstrates that REPAIR is the dominant factor—Figure 1 (both steps varied) and Figure 3 (only REPAIR varied) are nearly identical, while Figure 2 (only activation matching varied) shows much weaker effects.

### Variance Collapse as Explanatory Mechanism
Figure 4 provides a compelling explanation: accuracy barriers correlate strongly with variance ratio, and easy/hard data lead to systematically biased variance estimates. This mechanistic insight goes beyond just reporting "data matters."

### Comprehensive Experiments Across Architectures
The paper tests:
- Different widths (1×, 2×, 4×)
- Different normalization (BatchNorm, LayerNorm)
- Different scales (CIFAR-10, ImageNet)
- Results are consistent across settings, strengthening the conclusions

### Thorough Supplementary Material
The appendix provides detailed methodology, clear definitions of permutation symmetries, REPAIR, and EL2N scores, plus extensive additional results. This makes the work reproducible and accessible.

**Suggestions:**

### 1. Define EL2N Score in Main Text
The EL2N score (Paul et al., 2021) is central to the paper's methodology but is only defined in Appendix B.3. A brief definition in Section 2 would help readers understand what "easy" and "hard" mean without flipping to the supplement. Referencing to the formal definition in the appendix would be helpful too.

### 2. Provide Intuition for Easy Examples → Lower Variance
The finding that easy examples lead to underestimated variance (and thus larger barriers) is interesting but not intuitive. Why would easy examples produce lower activation variance? A brief explanation would strengthen the mechanistic story. Possible intuition: easy examples may produce more confident/peaked activations, underestimating the variance seen on the full distribution?

### 3. Explain LayerNorm Variance Ratio Patterns
In Figure 10 (LayerNorm), random data achieves variance ratio <1 while hard data is just above 1, yet both yield similar barriers. This seems to conflict with the "variance ratio ≈ 1 is optimal" story. A brief discussion of why LayerNorm behaves differently would be helpful.

### 4. Minor: Fix Broken References
There are missing/broken citations marked as "?" in the text:
- In Section B.1 (activation matching via OLS regression)
- In Section B.3 (EL2N score definition)

---

### Official Review · Reviewer_2Ehk · 2026-02-28

**Fit:** 2
**Significance:** 2
**Confidence:** 2

**Summary:**

They study how data choice affects model merging for independently trained ResNet models on image classification. They use a focusing on a two-stage model merging pipeline. First they match activations to align representations by permuting channels/neurons. Then they use REPAIR to correct post-merge activation/normalization statistics. Both stages consume data, and the paper varies the amount and type of data used for each stage to quantify sensitivity of the final merged model to data availability and distribution shift.

They operationalize easy versus hard examples via the EL2N difficulty score (Paul et al., 2021), computed per training example using early-training prediction error magnitude. They train multiple (10) ResNet20 models, compute EL2N at epoch 10, average across runs, and rank examples by this score. Using these ranked subsets for merging shows that data hardness can substantially affect merge quality; in particular, skewed subsets (especially easiest or hardest) can induce substantially larger (even 2x) accuracy barriers (difference in merged test set accuracy versus endpoint accuracy) than random subsets at the same data budget.

A central mechanism proposed for these effects is variance collapse, when the ratio between the variance (VR) of the merged models activations and the endpoint models is far from 1. The paper measures VR on a fixed probe distribution (a random 25\% subset of the training set), while REPAIR may be computed on different subsets (easy/hard/random), so distribution shift in the REPAIR subset can translate into VR mismatch on the probe and correlate with degraded test accuracy. This framing supports the practical conclusion that REPAIR is more sensitive to data representativeness than activation matching, and that biased subsets yield worse normalization/statistics calibration and worse merged performance.

**Strengths:**

The paper isolates the concrete and practically relevant question of what data is needed for successful model merging?, and answers it with a clean decomposition of the pipeline into two separable stages with different data dependencies. The separation between activation matching data and REPAIR data is particularly useful: the experiments provide evidence that, at a fixed budget, the data used for REPAIR has a disproportionately large effect on the final accuracy barrier, consistent with normalization/statistics being a fragile post-merge bottleneck.

The analysis in Section 3 connects performance to a measurable internal diagnostic (variance ratio), giving a mechanistic account of a common failure mode (variance collapse) rather than presenting only empirical curves. While the variance ratio $=1$ is optimal statement is idealized, the empirical relationship between VR drifting from $1$ and increased accuracy barriers provides a coherent explanation for why skewed, non-representative subsets (e.g., easiest by EL2N) can harm merging: they produce statistics that do not transfer to a representative probe distribution, and REPAIR calibrated on such subsets can therefore miscalibrate activation scales on typical inputs.

Finally, the paper is careful in how it defines difficulty and makes the easy/hard notion reproducible: EL2N is computed early in training and averaged across multiple seeds/models, avoiding a brittle single-run difficulty proxy. The result that difficulty-conditioned subsets can systematically degrade merging quality helps clarify what good data means in this context: representativeness for statistics estimation appears more critical than choosing examples that are individually informative or hard.

**Suggestions:**

A major limitation is that the difficulty-based selection does not appear to provide an actionable improvement over the strong baselines of random sampling; instead, the clearest prescription is use representative data for REPAIR, which random (or stratified-random) already approximates. As a result, the Section 3 story risks reading as primarily diagnostic: it offers an explanation why biased subsets fail (via variance mismatch and collapse) but does not turn that into a better data selection method at fixed budget. It could be interesting to explicitly frame subset selection as a moment-estimation problem and include constructive baselines designed to minimize discrepancy between subset-estimated statistics and full-distribution statistics. This would test whether the variance-collapse mechanism can be leveraged to outperform random, rather than merely rationalize why random is hard to beat.

Relatedly, the paper should be more explicit about evaluation distributions for diagnostics. In Figure 4(a), matching and REPAIR use the same subset, but the plotted variance ratio is still computed on a separate fixed probe (random 25\% of train), so VR need not be $1$ even when REPAIR strives to match moments; this subtlety is crucial for interpreting the easy/hard comparisons and should be highlighted prominently in the main text. More broadly, it would help to report VR measured both on the REPAIR subset and on the probe distribution, to disentangle REPAIR succeeds on its own data from REPAIR transfers to representative data and to quantify the distribution-shift effect directly.

Presentation can be improved. Figure 3 is difficult to interpret as currently written (the mapping from experimental condition to plotted quantity is not immediately clear), and several figures (e.g., 1--3) show results for three width settings without a clear definition of what width multiplier means operationally (e.g., channel scaling across all stages vs partial scaling) or why these specific multipliers were chosen. Since width materially changes overparameterization and thus merging behavior, the paper should define the width scaling precisely and summarize the qualitative differences across widths in the caption or surrounding text, rather than leaving the reader to infer that these correspond to different ResNet channel multipliers and repeated experiments across model sizes.

---

### Meta-Review · Area_Chair_VLiZ · 2026-03-01

**Recommendation:** Accept

**Metareview:**

The paper is well written, easy to understand, and tackles an important and timely research question with interesting findings. I recommend acceptance.

---

### Decision · Program_Chairs · 2026-03-02

Accept